# *Capsella bursa-pastoris* Is a Key Overwintering Plant for Aphids in the Mediterranean Region

**DOI:** 10.3390/insects12080744

**Published:** 2021-08-18

**Authors:** Serdar Satar, Nickolas G. Kavallieratos, Mustafa Tüfekli, Gül Satar, Christos G. Athanassiou, Nikos E. Papanikolaou, Mehmet Karacaoğlu, Işıl Özdemir, Petr Starý

**Affiliations:** 1Department of Plant Protection, Faculty of Agriculture, Çukurova University, Adana 01330, Turkey; 2Laboratory of Agricultural Entomology and Zoology, Department of Crop Science, Agricultural University of Athens, 75 Iera Odos str., 11855 Athens, Greece; nick_kaval@aua.gr (N.G.K.); nikosp@aua.gr (N.E.P.); 3Biological Control Research Institute, Kışla str., Adana 01321, Turkey; m.tufekli@bmi.gov.tr; 4Biotechnology Research and Application Center, Çukurova University, Adana 01330, Turkey; satarg@cu.edu.tr; 5Laboratory of Entomology and Agricultural Zoology, Department of Agriculture, Crop Production and Rural Environment, University of Thessaly, Phytokou str., 38446 Nea Ionia, Greece; athanassiou@agr.uth.gr; 6Directorate of Plant Produce Protection, Greek Ministry of Rural Development and Food, 150 Sygrou Ave., 17671 Athens, Greece; 7Department of Plant Protection, Faculty of Agriculture, Turgut Özal University, Malatya 44000, Turkey; mehmet.karacaoglu@ozal.edu.tr; 8Directorate of Plant Protection Central Research Institute, Gayret Mahallesi, Fatih Sultan Mehmet Bulvarı, No 66, Yenimahalle, Ankara 06172, Turkey; isil.ozdemir@tarimorman.gov.tr; 9Institute of Entomology, Biology Centre, Laboratory of Aphidology, AVCR, Branišovska 31, 37005 České Budějovice, Czech Republic; stary@entu.cas.cz

**Keywords:** Aphididae, Aphidiinae, hyperparasitoids, shepherd’s purse, trophic relationships

## Abstract

**Simple Summary:**

Reservoir plants are crucial for overwintering pests and their biological control agents. A long-term survey revealed that *Capsella bursa-pastoris* is a significant host plant, especially for aphids as well as their parasitoids and hyperparasitoids. Twenty-five aphids and eleven parasitoid species were identified on this weed. *Myzus persicae* and *Aphis gossypii* were the most commonly recorded aphid species, and *Binodoxys angelicae* the most frequent parasitoid. Additionally, the monthly distribution of the aphids, parasitoids, and hyperparasitoids showed that *C. bursa pastoris* fills the host plant gap in the absence of crops. Trophic relations within the community and the importance of *C. bursa pastoris* were also analyzed in this study.

**Abstract:**

The reproduction of aphids depends to a great extent on their host plants, an integration that impacts on the successful expansion of overwintering populations. Therefore, a survey was conducted to evaluate the globally distributed *Capsella bursa-pastoris* as an overwintering host of economically important aphid species, their parasitoids and hyperparasitoids in the southern and western regions of Turkey from November to March in 2006 to 2013. During this survey, 395 samples of *C. bursa-pastoris* were collected with 25 aphid species recorded. Among aphids that feed on this host, *Myzus persicae*, *Aphis gossypii*, *Rhopalosiphum padi*, *Aphis fabae*, *Aphis craccivora*, *Lipaphis erysimi*, and *Brevicoryne brassicae* were the most frequently recorded. In total, 10,761 individual parasitoids were identified. *Binodoxys angelicae*, *Aphidius colemani*, *Aphidius matricariae*, *Diaeretiella rapae*, *Ephedrus persicae*, and *Lysiphlebus confusus* were the most abundant aphidiines that emerged from the aphids collected from *C. bursa-pastoris*. *Alloxysta* spp. (Hymenoptera: Cynipoidea), Chalcidoidea (unidentified at genus level), and *Dendrocerus* spp. (Hymenoptera: Ceraphronoidea) were identified as hyperparasitoids on the parasitoids. These findings indicate that *C. bursa-pastoris* is a key non-agricultural plant that significantly contributes to the overwintering of numerous aphids and their parasitoids, which should be given serious consideration when biological control strategies are designed.

## 1. Introduction

*Capsella bursa-pastoris* (L.) Medik. (Brassicales: Brassicaceae) is globally distributed, being adapted to cool, temperate and subtropical climates. It can survive under high or low temperatures, to as low as −12 °C, and at different elevations, i.e., from the sea level to the high Himalayas [1]. This species is observed from autumn to late spring in Mediterranean climates [2]. The center of origin of *C. bursa-pastoris* is considered to be in Anatolia; it is an historical plant in daily life, and possible records date back to 5850–5600 BC [3]. It hosts many insect species in the Coleoptera, Diptera, Hemiptera, Hymenoptera and Lepidoptera, as well as other invertebrates in the Arachnida and Mollusca [1]. In the Aphididae (Hemiptera), over 40 species have been identified as feeding on this host [4]. *C. bursa-pastoris* is also a host for beet mild yellowing, beet western yellowing, potato spotted wilt, and potato leafroll viruses [5,6]. In addition, being a host of so many aphid species further increases the importance of this plant with respect to carrying and spreading the viruses to the agriculture plants.

Aphids are considered as one of the most important groups of plant pests in Turkey, as in most countries [4,7,8,9]. The aphid fauna around the world includes about 5000 species belonging to 510 genera [4], with over 500 species identified in different geographic and climatic regions of Turkey [10,11]. Aphids cause direct and indirect damage, including loss of sap, deformities, changes in color, abnormal development, and reduction in photosynthesis due to sooty mold growth. In addition, over 200 aphid species are known as vectors of some 300 viruses globally, and aphids represent about half of all insect vectors of viruses [12].

Integrated pest management (IPM) strategies endeavor to combine different methods in order to sufficiently decrease aphid numbers [13]. Biological control, as an indispensable part of IPM strategies, includes the use of Aphidiinae parasitoids against aphids, as effective natural enemies [14,15]. Aphidiines are solitary endophagous parasitoids exclusive to aphids [13,16]. Currently, the available information on the Aphidiinae occurring in Turkey and their aphid–plant associations mostly comes from general surveys of various agricultural and non-agricultural plants [17,18,19,20,21,22,23,24,25,26]. However, the diversity and usefulness of parasitoids in Turkey could be enhanced by the further development of release techniques and habitat management strategies. Various plants that occur near cultivated fields may be alternative hosts for phytophagous species and natural enemies extending beyond the production season [27,28]. Although non-agricultural plants, both annuals and perennials, can be sources of pests, they can also be valuable sources of important natural biological control agents [29,30]. Many researchers have studied the trophic relationships among non-agricultural plants and their aphids and parasitoids, including on *Cirsium arvense* (L.) Scop. [31], *Philadelphus coronarius* L. [32], *Chenopodium* weeds [33], midfield thickets [34], *Dittrichia viscosa* (L.) Greuter and *Rubus ulmifolius* Schott [35], *Salix* spp. and *Populus* spp. [36], *Vitex agnus-castus* L. and *Euphorbia characias* ssp. *wilfenii* (Hoppe ex W.D.J.Koch) Radcl.-Sm. [37], *Hieracium* spp. [38] and ornamental plants [39]. Such research has provided a better understanding of the highly complex relationships within these communities and the basis for the development of management strategies.

Several species of anholocyclic aphids overwinter under Mediterranean climatic conditions, and therefore the knowledge about their overwintering host plants is critical for forecasting spring infestations [40]. However, there appears to be no information on the role of *C. bursa-pastoris* as a reservoir for parasitoids of overwintering aphids. Therefore, the objectives of this research were to determine (1) the importance of *C. bursa-pastoris* as a winter host for aphid species and their associated parasitoids and hyperparasitoids, (2) trophic relationships among these species, (3) interactions between aphid and parasitoid species, and between parasitoids and hyperparasitoids, and (4) the monthly distribution of the species in this community during the presence of *C. bursa-pastoris* over an extended period of sampling in southern and western Turkey. 

## 2. Materials and Methods

### 2.1. Sampling Area

*Capsella bursa-pastoris* samples (n = 395) were randomly collected from orchards and non-cropping areas from November to March in 2006 (n = 16), 2007 (n = 102), 2008 (n = 111), 2009 (n = 99), 2011 (n = 15), 2012 (n = 47) and 2013 (n = 5) from southern (Adana, Antalya, Hatay, Kahramanmaraş, Mersin, and Osmaniye), and western (İzmir and Muğla) provinces in Turkey. During the sampling period, since the region was (and remains) the main citrus production area of Turkey, the sampling was mainly conducted in and around citrus orchards.

### 2.2. Sampling

Samples of *C. bursa-pastoris* infested with aphids and aphid mummies were collected from field locations. Adult aphids were preserved in 90% ethanol and 75% lactic acid at 2:1 [41] for identification, and parasitoids and hyperparasitoids were reared from mummies in the laboratory. The plant samples were placed in plastic containers (5 L), separating aphid species, covered with muslin and placed in a growth room at 22 °C, 65% RH and 16:8 h L:D photoperiod [37]. The containers were inspected daily for emerged parasitoids and hyperparasitoids, which were collected with an aspirator, and killed in 96% ethanol. The specimens were identified by Petr Starý and Nickolas G. Kavallieratos. Some specimens were point- or slide-mounted for detailed examination. For the dissection or whole mounting, the specimens were boiled in 10% KOH for 2 min after washing in water, then rewashed and placed onto the slide in a drop of Faure–Berlese medium [42]. Olympus SZX9 (Olympus Corporation, Tokyo, Japan) or SMXX Carl Zeiss Jena (Carl Zeiss MicroImaging GmbH, Göttingen, Germany) stereomicroscopes were used for the examination of the external morphology. Voucher specimens were deposited in the collection of P. Starý at České Budějovice (Czech Republic) and in the collection of the Laboratory of Agricultural Zoology and Entomology (Greece).

### 2.3. Data Analysis

Statistical analysis was performed to examine (1) the abundance of the most commonly detected parasitoid species on *Aphis craccivora* Koch, *Aphis fabae* Scopoli, *Aphis gossypii* Glover, *Aphis nasturtii* Kaltenbach, *Brevicoryne brassicae* (L.), *Lipaphis erysimi* (Kaltenbach), *Myzus persicae* (Sulzer) and *Rhopalosiphum padi* (L.), and (2) the associations of *Aphidius colemani* Viereck, *Aphidius matricariae* Haliday, *Binodoxys angelicae* (Haliday), *Diaeretiella rapae* (M’Intosh), *Ephedrus persicae* Froggatt and *Lysiphlebus confusus* Tremblay and Eady with the most commonly detected aphids using the chi-square test [43] with the SPSS 17.0 (IBM, Armonk, NY, USA). A range of models were analyzed (SPSS 17.0) for both datasets ((1) and (2)) using generalized linear models (GLM) applicable for repeated measures to test the interaction between aphid and parasitoid species. Aphid species was the dependent variable and the number of each parasitoid was the fixed factor for the first GLM analysis; parasitoid species was the dependent variable and the number of each parasitoid in each sampled aphid was the fixed factor for the second GLM analysis.

## 3. Results

*Capsella bursa-pastoris* was found to host 25 aphid and 12 parasitoid species, and individuals from three families of hyperparasitoids (Figure 1, Figure 2, Figure 3 and Figure 4). The aphids identified to genera or species level were *Acyrthosiphon pisum* (Harris), *Aphis* sp., *A. craccivora, A. fabae, Aphis fabae solanella* Theobald, *A. gossypii*, *A. nasturtii*, *Aulacorthum solani* (Kaltenbach), *Capitophorus* sp., *Cavariella* sp., *Coloradoa* sp., *Brachycaudus cardui* (L.), *Brachycaudus helichrysi* (Kaltenbach), *B. brassicae, Hayhurstia atriplicis* (L.), *Hyadaphis foeniculi* (Passerini), *Hyperomyzus lactucae* (L.), *L. erysimi*, *Macrosiphum euphorbiae* (Thomas), *Myzaphis rosarum* (Kaltenbach), *Myzus cerasi* (F.), *Myzus ornatus* Laing, *M. persicae*, *R. padi* and *Rhopalosiphum maidis* (Fitch). *M. persicae* was the most common species, at 28% of all aphid specimens, followed by *A. gossypii* (24%) and *R. padi* (9%) (Figure 1). Eleven parasitoid taxa were identified from 10,761 specimens, viz. *Aphelinus* sp. (Hymenoptera: Aphelinidae), *A. colemani*, *A. matricariae*, *A. transcaspicus* Telenga, *B. angelicae*, *D. rapae*, *Ephedrus nacheri* Quilis, *E. persicae*, *Lysiphlebus fabarum* (Marshall), *Lysiphlebus testaceipes* (Cresson) and *Praon volucre* (Haliday) (Hymenoptera: Braconidae: Aphidiinae) (Figure 2). *B. angelicae* was the most common, at 58% of all parasitoids, followed by *A. colemani* and *A. matricariae*. *M. persicae*, *A. gossypii* and *A. fabae* were parasitized with 10, 10 and 8 parasitoid species, respectively. Conversely, *B. angelicae*, *A. colemani* and *A. matricariae* parasitized 14, 12 and 11 aphid species, respectively (Figure 3). Seven genera were identified among 477 specimens. The hyperparasitoids obtained from 101 samples were generally from mixed parasitoid populations. *Alloxysta* spp. (Hymenoptera: Cynipoidea) were the most commonly (n = 298) identified hyperparasitoids, followed by Chalcidoidea (unidentified at genus level) (n = 73), and *Dendrocerus* spp. (Hymenoptera: Ceraphronoidea) (n = 43). *Alloxysta* spp. and chalcid wasps were determined from parasitoids on 10 and 7 aphid species, respectively. All hyperparasitoids were identified from host parasitoids on *A. fabae*, *A. gossypii* and *M. persicae* (Figure 3 and Figure 4).

Figure 4 shows the parasitoid and hyperparasitoid population composition, relative proportions and number of collections for each host aphid species. Some aphid species and their parasitoids or hyperparasitoids with only a few detections (in one to two samples) were omitted from this evaluation. *B. angelicae* was the major parasitoid for the common aphid species on *C. bursa-pastoris*, viz. *M. persicae*, *A. craccivora*, *A. gossypii*, and *A. fabae*. In addition, *D. rapae* on *B. brassicae*, and *A. colemani* on *L. erysimi* were the predominant parasitoid species. *R. padi* had a different parasitoid composition than others, with close ratios of *B. angelicae*, *L. confusus* and *A. matricariae*. *Alloxysta* spp. were detected predominantly as the parasitoids of *A. gossypii*, *L. erysimi*, *M. persicae* and *R padi*, and were also collected as parasitoids of *A. fabae*. However, high parasitoid numbers but few hyperparasitoids were obtained from *A. craccivora*, *A. nasturtii* and *B. brassicae*.

Mixed aphid populations were found in 19% of samples with aphids (Figure 1). About 74% of parasitoids (n = 8005) were obtained from only one aphid host species (n = 17), about 22% (n = 2316) from two hosts (n = 21), 3.4% (n = 365) from three hosts (n = 9) and 0.7% (n = 75) from four hosts (n = 2). Almost half of the parasitoids were observed in combinations with *A. fabae*, followed by *A. gossypii* (Table 1). *A. fabae*, *M. persicae* and *A. gossypii,* with 13, 7 and 6 aphid combinations, respectively. *A. colemani*, *A. matricariae* and *B. angelicae* were the most common parasitoid species from these aphid combinations.

### Relationship between Aphids and Parasitoids

The chi-square analysis indicated significant differences in the abundance of the parasitoids of *A. craccivora*, *A. fabae*, *A. gossypii*, *A. nasturtii*, *B. brassicae*, *L. erysimi*, *M. persicae* and *R. padi*. In addition, the analysis showed significant differences in aphid associations with parasitoids species: *A. colemani*, *A. matricariae*, *B. angelicae*, *D. rapae*, *E. persicae* and *L. confusus* (Table 2). GLM analysis revealed that parasitoid abundance in aphid species was significant (df_aphid_ = 7, 519, F_aphid_ = 4.62, p_aphid_ < 0.01; df_parasitoid_ = 5, 519, F_parasitoid_ = 6.37, p_parasitoid_ < 0.01; df_aphid*parasitoid_ = 29, 519; F_aphid*parasitoid_ = 3.26; p_aphid*parasitoid_ < 0.01); also, aphid preference among parasitoids was significant (df_parasitoid_ = 5, 519, F_parasitoid_ = 5.98, *p* < 0.01; df_aphid_ = 7, 519, F_aphid_ = 4.94, *p* < 0.01; df_parasitoid*aphid_ = 27, 519, F_parasitoid*aphid_ = 3.49, *p* < 0.01).

Sampling of *C. bursa-pastoris* was conducted from November to March in 2006 to 2013 (Figure 5) and different aphid species were found in different months. *A. fabae*, *B. brassicae*, *R. padi*, *M. persicae* and *L. erysimi* were most abundant in March, *A. gossypii* and *A. nasturtii* in February, and *A. craccivora* in December. Parasitoid species composition and numbers differed between months. *L. confusus* and *L. fabarum* were most abundant in January. *A. matricariae*, *A. colemani*, *E. persicae* and *P. volucre* were most abundant in February, followed by March. Conversely, *B. angelicae* had the highest numbers in March, followed by February and December. The highest frequency of aphids and numbers of parasitoid and hyperparasitoid were recorded in March, followed by April, with the exception of some hyperparasitoids, with numbers peaking in December. 

## 4. Discussion

Non-agricultural plants can provide food sources for both pests and beneficial organisms, acting as alternative food sources for pests in the absence of crops and as hosts of alternative prey for beneficial organisms [38,44]. In this respect, the present research elucidates the trophic associations among host plants, aphids, parasitoids and hyperparasitoids, and the importance of *C. bursa-pastoris* as an overwintering host for these communities. The vast majority of the aphid species recorded in this survey are included in the lists of Aksoy et al. [1] and Blackman and Eastop [4]. However, *Capitophorus* sp., *Cavariella* sp., *Coloradoa* sp., *H. atriplicis*, *H. foeniculi*, *H. lactucae*, *M. rosarum*, *M. ornatus* and *R. maidis* are not included in Blackman and Eastop [4]. These new findings enhance the importance of *C. bursa-pastoris* as a reservoir plant for aphids and provide clear evidence that this plant can host many more aphid species than initially recognized. 

The host associations of aphids depend on primary and secondary metabolites, and host plant characters [45]. For example, *M. persicae*, the most abundant species on *C. bursa-pastoris* in the present study, has shorter growing and doubling time on *C. bursa-pastoris* in relation to other plant species belonging to the same or different botanical families including *Chenopodium album* L., *Amaranthus retroflexus* L. (Amaranthaceae), *Cardaria draba* (L.) Desv., *Lepidium perfoliatum* L., *Raphanus sativus* L. (Brassicaceae), *Convolvulus arvensis* L. (Convolvulaceae) and *Solanum sarrachoides* Sendtn. (Solanaceae) [46]. Glucosinolates as secondary plant compounds have been associated with the defense mechanisms of brassicaceous plants against *M. persicae* [47]. In addition, the nutritional status of the host plant is also important for *M. persicae*—for example, higher levels of secondary plant metabolites than those on susceptible cultivars can reduce the fitness of the aphid [48]. The morphological characteristics of *C. bursa-pastoris* are another factor contributing to the wide range of aphid species feeding on this plant. The basal leaves of *C. bursa-pastoris* form a rosette that is close to the soil, providing a sheltered environment for aphids and natural enemies during the winter. In addition, its flowers in spring provide food for adult parasitoids. Nectar and pollen increase the fecundity and life span of the beneficial insects [44]. Hence, the flowers of *C. bursa-pastoris* offer an alternative food source for parasitoid adults, and may help to improve their reproduction and rates of parasitism [49,50]. 

Sampling of *C. bursa-pastoris* in the present study was conducted mostly near the citrus production areas (376 of 484 samples). Although *M. persicae* was found to be the major aphid species infesting this host, it is only a minor pest of citrus [49]. However, *M. persicae* has a broad host range, including *Citrus* spp., *Prunus* spp., solanaceous crops, and many wild annual and perennial plants in the sampled region [10,51]. Stone-fruit orchards, which represent the major hosts for *M. persicae*, are increasing in number in the region because farmers prefer these early maturating crops in the Mediterranean climate for their cash flow benefits. Since, worldwide, *M. persicae* is the primary vector of plum pox virus (sharka) [51], *C. bursa-pastoris* needs to be considered in developing vector control strategies. 

*A. gossypii* also occurred at a high frequency on *C. bursa-pastoris*, and is one of the two major aphid species on citrus [49]. Hence, *C. bursa-pastoris* is probably more directly involved in the trophic association between *A. gossypii* and citrus than in *M. persicae* and citrus. The rapid increase in air temperature in early spring makes *A. gossypii* and *Aphis spiraecola* Patch more suitable than *M. persicae* due to their reproductive potential at high temperature and because of the spread to larger areas in a short timeframe [7,50]. Studies on the effect of secondary metabolites are required to clarify the relationship between *C. bursa-pastoris*, citrus, and *A. gossypii* or *M. persicae.*


The diversity of parasitoids and their population densities on non-agricultural plants contribute to their function as biological control agents in crops [52]. Satar et al. [25] reported that *B. angelicae* was the major parasitoid of *A. gossypii* feeding on citrus in the region. Since this parasitoid was recorded in high numbers from this aphid on *C. bursa-pastoris*, it is hypothesized that populations of *B. angelicae* move between citrus orchards and *C. bursa-pastoris*, which would positively contribute to natural biological control [38,52]. In addition, *C. bursa-pastoris* hosts aphids which can support parasitoid and hyperparasitoid survival during the unfavorable winter and early spring months when there are no aphids in the cropping systems of the surveyed area. The presence of parasitoids on *C. bursa-pastoris* could be critical to suppress other common aphid species including *A. craccivora*, *A. fabae* and *M. persicae*.

The abundance of parasitoids of different aphids varied significantly and supported by GLM analysis. However, the parasitoid abundance associated with the genus *Aphis* was similar to other genera. The greatest difference was between *B. brassicae* and *M. persicae*, even though both are major crucifer aphids. *Brevicoryne brassicae* has many alternative winter hosts, including cabbage cauliflower, broccoli, radish and wild mustard [53], which might be more suitable hosts than *C. bursa-pastoris*. The aphid associations of *A. colemani*, *A. matricariae*, *B. angelicae*, *D. rapae*, *E. persicae* and *L. confusus* were statistically significant, but *E. persicae* had the smallest χ^2^ value compared to the other parasitoids. The abundance of *E. persicae* in association with *A. matricariae*, *B. angelicae* and *A. colemani* on the three most common aphids, *M. persicae*, *A. gossypii* and *R. padi*, respectively, had a more balanced distribution compared to the other parasitoids. 

Differences in host associations could be attributed to olfactory co-effects of host plants and aphids towards aphid parasitoids. In the absence of the host plant, the parasitoids are limited or not attracted by aphids [48]. The parasitism rate of *Aphidius ervi* Haliday and *Aphidius rhopalosiphi* De Stefani were significantly stimulated by the presence of both aphids and host plants [54]. Volatiles of plants attract the parasitoids in the absence of the host aphid. For example, *A. rhopalosiphi* was attracted by vermiculite impregnated with wheat extract [55]. In addition, the metabolites of some plants prevent parasitoids parasitizing aphids. Vinson [56] reported that *Aphidius smithi* Sharma and Subba Rao could be reared on *M. persicae* on broad bean but not on tobacco. Aphid sucking causes the release of volatiles from plants as a defense response. The parasitoids are also attracted by aphid-induced plant volatiles [57,58]. Kos et al. [59] showed that higher glucosinolate concentrations in *Arabidopsis thaliaca* (L.) Heynh. ecotypes infested with *B. brassicae* resulted in larger aphids, which was in turn positively correlated with *D. rapae* performance. Another important feature of *C. bursa-pastoris* is it having mixed aphid populations, as it is known to host over 40 aphid species [4]. Infested *C. bursa-pastoris* produces kairomone (help signal), which attracts parasitoids [60], but because *C. bursa-pastoris* can be infested with up to five aphid species, this means that some aphids present might not be suitable hosts for the attracted parasitoids. This mixed population can reduce parasitoid activity because some parasitoids are monophages or oligophages.

One of the most important findings of the present study is the overall low activity of hyperparasitoids (<5%) associated with parasitoids of aphids feeding on *C. bursa-pastoris*. Given that hyperparasitoids are considered to negatively affect biological control practice [61], their low numbers in parasitoids of aphids feeding on *C. bursa-pastoris* is a noteworthy finding. When aphids are parasitized on the shoots of plants, they move to sheltered locations to protect others on the plant [62,63], and the basal leaves of *C. bursa-pastoris* provide a sheltered location for parasitized individuals.

The population density of the parasitoids changed over the course of the year. The higher activity of different species in different months is important for the suppression of aphid densities. Aphid activity on citrus starts in early March and continues until December in the surveyed region [49]. However, parasitoid activity occurred mostly in April to June and October to December [25]. The life cycle of *C. bursa-pastoris* in citrus orchards fills the gap in the absence of aphids during the winter months and serves to support parasitoid dispersal to citrus trees in March, when aphid populations start to develop in the citrus. Hyperparasitoids peaked in spring, and peaked again on overwintering parasitoids in December, indicating that they can find suitable hosts for overwintering. Although hyperparasitoids were considered harmful to the primary parasitoids, they have positive ecological effects helping to maintain the natural balance in insect populations [61]. Thus, the knowledge about hyperparasitoids is important to understand their interactions in the insect communities, especially in how this impacts biocontrol strategies.

The results presented here indicate that *C. bursa-pastoris* is of vital importance for the anholocyclic life cycle of aphids in the Mediterranean Basin. More than 500 aphid species have been recorded in Turkey [10,11], and in this research, 25 of these species were found to have parthenogenetic reproduction on *C. bursa-pastoris* in the Mediterranean Region of Turkey. Spring populations of aphids on vegetables, citrus and stone fruits are related to *C. bursa-pastoris* supporting aphid populations. Nevertheless, *C. bursa-pastoris* is important because it also contributes to the persistence of parasitoid and hyperparasitoid diversity in the region by hosting a remarkably high number of these insects over long periods. Therefore, *C. bursa-pastoris* should be seriously considered during the development of management strategies for Mediterranean agricultural systems.

## Figures and Tables

**Figure 1 insects-12-00744-f001:**
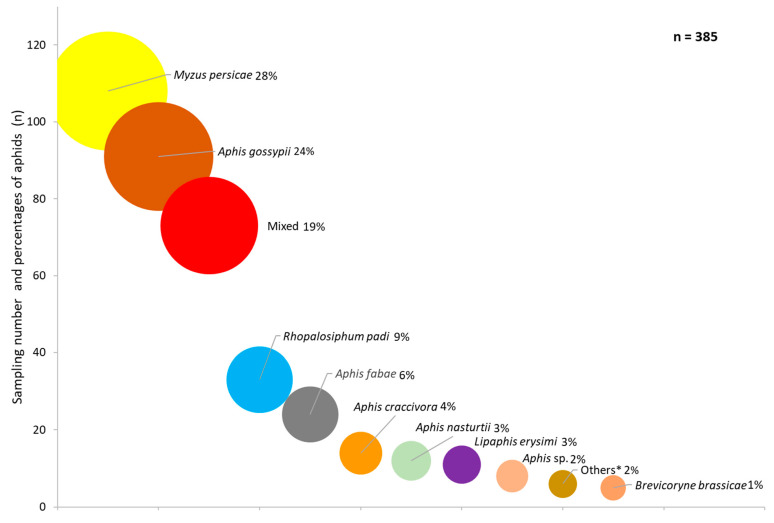
Sampling number and percentages of aphid species on *Capsella bursa-pastoris* in southern and western Turkey from November to March in 2006 to 2013 (* *Aulacorthum solani*, *Brachycaudus cardui*, *Brachycaudus helichrysi*, *Capitophors* sp., *Hyadaphis foeniculi*, *Myzus cerasi*).

**Figure 2 insects-12-00744-f002:**
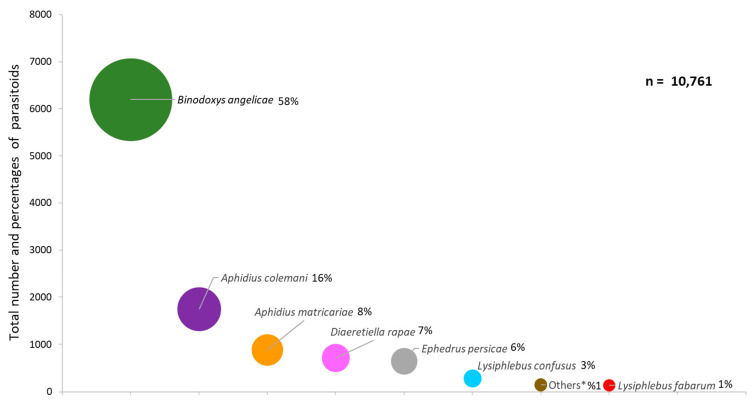
Total number and percentages of parasitoid species occurring on *Capsella bursa-pastoris* in southern and western Turkey from November to March in 2006 to 2013 (* *Aphelinus* sp., *Aphidius transcaspicus*, *Ephedrus nacheri*, *Lysiphlebus testaceipes*, *Praon volucre*).

**Figure 3 insects-12-00744-f003:**
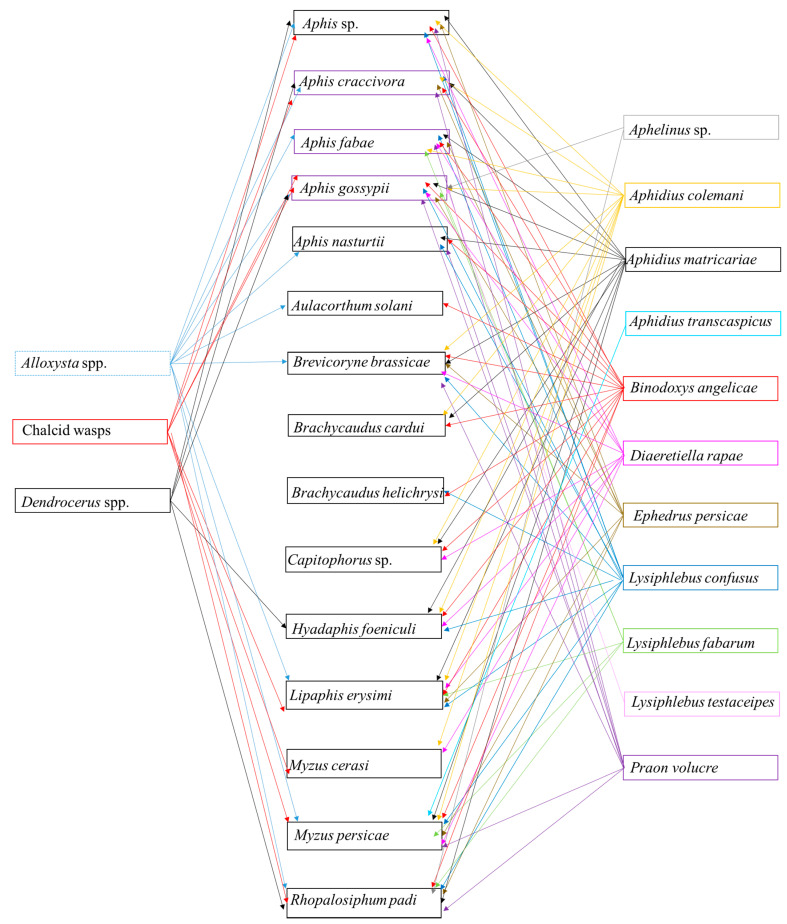
Trophic relationships among aphids, parasitoids and hyperparasitoids occurring on *Capsella bursa-pastoris* in southern and western Turkey from November to March in 2006 to 2013.

**Figure 4 insects-12-00744-f004:**
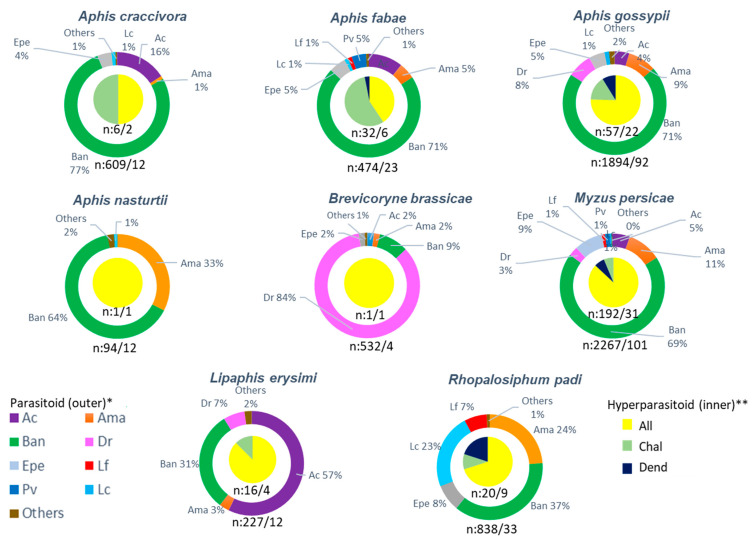
Percentages and total number of the parasitoid/sampling number (outer circle) and hyperparasitoid/sampling number (inner circle) for different aphid species feeding on *Capsella bursa-pastoris*. Abbreviations-Parasitoids *: *Aphidius colemani* (Ac), *Aphidius matricariae* (Ama), *Binodoxys angelicae* (Ban), *Diaeretiella rapae* (Dr), *Ephedrus persicae* (Epe), *Lysiphlebus confusus* (Lc), *Lysiphlebus fabarum* (Lf), *Praon volucre* (Pv). Hyperparasitoids **: *Alloxysta* spp. (All), Chalcid wasps (Chal), *Dendrocerus* spp. (Dend).

**Figure 5 insects-12-00744-f005:**
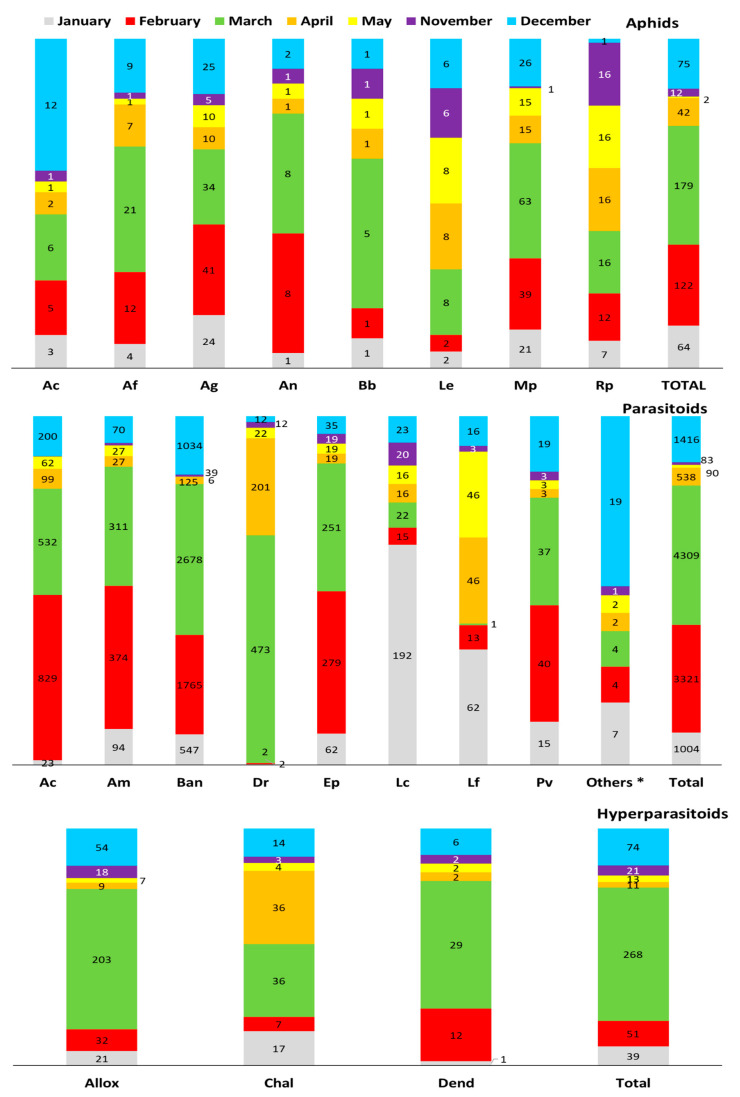
Seasonal abundance of aphids, parasitoids and hyperparasitoids occurring on *Capsella bursa-pastoris* in southern and western Turkey between 2006 and 2013 (*Rhopalosiphum padi* (Rp); *Myzus persicae* (Mp); *Lipaphis erysimi* (Le); *Brevicoryne brassicae* (Bb);*Aphis nasturtii* (An); *Aphis gossypii* (Ag); *Aphis fabae* (Af); *Aphis craccivora* (Ac); *Praon volucre* (Pv); *Lysiphlebus fabarum* (Lf); *Lysiphlebus confusus* (Lc); *Ephedrus persicae* (Ep); *Diaeretiella rapae* (Dr); *Binodoxys angelicae* (Ban); *Aphidius matricariae* (Ama); *Aphidius colemani* (Ac), *Dendrocerus* spp. (Dend); Others * (*Aphelinus* sp.; *Aphidius transcaspicus*; *Ephedrus nacheri*; *Lysiphlebus testaceipes*); chalcid wasps (Chal); *Alloxysta* spp. (Allox).

**Table 1 insects-12-00744-t001:** Aphidiinae parasitoid species and their numbers on mixed aphid populations occurring on *Capsella bursa-pastoris* in southern and western Turkey from November to March in 2006 to 2013.

Main Aphid Species	Mixed Aphid Species	Parasitoid Species				
		*A. colemani*	*A. matricariae*	*B. angelicae*	Others	Total
*A.* *fabae*	*A. pisum, Aphis* sp.,	953	38	564	133	1688
	*A. craccivora, A. fabae solanella, A. gossypii, Cavariella* sp., *Coloradoa* sp., *H. foeniculi,*					
	*H. lactucae, L. erysimi,*					
	*M. ornatus, M. persicae,*					
	*M. rosarum*					
*A. gossypii*	*A. craccivora, B. brassicae,*	41	106	397	165	709
	*M. persicae, Phopalosiphum* sp., *R. maidis*					
*B. brassicae*	*Capitophorus* sp., *M. persicae*	12	0	8	2	22
*Hyadaphis* sp.	*L. erysimi*	2	2	23	0	27
*M. persicae*	*Aphis* sp., *A. craccivora,*	3	19	43	14	79
	*R. maidis, R. padi*					
*R. padi*	*A. craccivora, A. fabae solanella*	1	0	3	3	7

**Table 2 insects-12-00744-t002:** Differences in the abundances of parasitoids found in aphids, and aphid–parasitoid associations on *Capsella bursa-pastoris* in southern and western Turkey from November to March in 2006 to 2013 (for all cases *p* < 0. 01).

Parasitoid Numbers in Aphid Species	Df	χ^2^	Aphid-Parasitoid Associations	Df	χ^2^
*A. craccivora* vs. *A. fabae*	5	27.9	*A. colemani* vs. *A. matricariae*	7	434.5
*A. craccivora* vs. *A. gossypii*	5	169.4	*A. colemani* vs. *B. angelicae*	7	498.1
*A. craccivora* vs. *A. nasturtii*	4	180.4	*A. colemani* vs. *D. rapae*	5	640.9
*A. craccivora* vs. *B. brassicae*	5	923.5	*A. colemani* vs. *E. persicae*	6	238.4
*A. craccivora* vs. *L. erysimi*	5	188.6	*A. colemani* vs. *L. confusus*	6	503.1
*A. craccivora* vs. *M. persicae*	5	205.4			
*A. craccivora* vs. *R. padi*	4	491.1			
*A. fabae* vs. *A. gossypii*	5	59.7	*A. matricariae* vs. *B. angelicae*	7	384.6
*A. fabae* vs. *A. nasturtii*	5	73. 7	*A. matricariae* vs. *D. rapae*	7	793.5
*A. fabae* vs. *B. brassicae*	5	755.9	*A. matricariae* vs. *E. persicae*	7	68.8
*A. fabae* vs. *L. erysimi*	5	184.5	*A. matricariae* vs. *L. confusus*	7	196.9
*A. fabae* vs. *M. persicae*	5	88.5			
*A. fabae* vs. *R. padi*	5	317.8			
*A. gossypii* vs. *A. nasturtii*	5	69.7	*B. angelicae* vs. *D. rapae*	7	2345.3
*A. gossypii* vs. *B. brassicae*	5	1399.6	*B. angelicae* vs. *E. persicae*	7	185.0
*A. gossypii* vs. *L. erysimi*	5	504.9	*B. angelicae* vs. *L. confusus*	7	910.5
*A. gossypii* vs. *M. persicae*	5	181.9			
*A. gossypii* vs. *R. padi*	5	675.8			
*A. nasturtii* vs. *B. brassicae*	5	413.2	*D. rapae* vs. *E. persicae*	6	636.2
*A. nasturtii* vs. *L. erysimi*	5	118.9	*D. rapae* vs. *L. confusus*	6	715.7
*A. nasturtii* vs. *M. persicae*	5	40.7			
*A. nasturtii* vs. *R. padi*	3	43.6			
*B. brassicae* vs. *L. erysimi*	5	448.8	*E. persicae* vs. *L. confusus*	6	241.8
*B. brassicae* vs. *M. persicae*	5	1373.7			
*B. brassicae* vs. *R. padi*	5	1087.9			
*L. erysimi* vs. *M. persicae*	5	307.0			
*L. erysimi* vs. *R. padi*	5	585.8			
*M. persicae* vs. *R. padi*	5	458.4			

## Data Availability

Data supporting the information shown in the results have been uploaded to Mendeley Data. Accessed date: 12 May 2021 http://dx.doi.org/10.17632/c5334rbvh5.1.

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
