# Peer review of "Capsella bursa-pastoris Is a Key Overwintering Plant for Aphids in the Mediterranean Region"

_insects, 2021, doi:10.3390/insects12080744_

Round 1
Reviewer 1 Report
Comments to the Author
The paper describes the survey of aphids and parasitoid species on Capsella bursa-pastoris in the mediterranean region. However, this survey was conducted eight years ago, how to make sure it reflects the current situation. For data analysis, the description concerns statistical analysis need be more detailed. It's better to provide one geographic distribution maps, that will be easy to help readers to understand the sampling area.
Author Response
Dear referee,
Thank you for your editing and comments. We did all your corrections to the text. We add a sentence about sampling area profile in today. We did GLM analyses and add to text.
Best regards

Reviewer 2 Report
The paper "Capsella bursa-pastoris is a key overwintering plant for aphids in the Mediterranean Region" is interesting and worth publishing. I only underlined mispelling of some verbs or scientific names I highlighted in the attached file.

Author Response
Dear referee,
Thank you for your editing and comments. We made all your corrections to the text.
Best regards

Reviewer 3 Report
The manuscript ‘Capsella bursa-pastoris is a key overwintering plant for aphids in the Mediterranean Region’ by Satar et al. reports on a 2006-2013 survey studying the aphid, parasitoid and hyper-parasitoid diversity found on Capsella bursa-pastoris in Southern and Western regions of Turkey, with a focus on the role of the plant as a reservoir for overwintering insects. The manuscript is an interesting study into the diversity and trophic interactions of aphids, their parasitoids and hyper-parasitoids on one specific host plant.
Unfortunately, the manuscript has not been formatted with line numbering, making it difficult to provide detailed comments. In the absence of line numbering, I will have to keep my comments broad.
1) The level of English needs to be improved. For example, the Simple Summary contains many phrases that do not make sense in English including:
‘as parasitoid come forward’
‘in the condition of absence’
There are also errors throughout, such as the opening of the Introduction:
‘Capsella bursa-pastoris has is globally distributed’
And the opening of the Discussion:
‘Non-agricultural plants can be provide food sources’
2) I feel the Materials and Methods could benefit on more detail of the sampling methods used. For example, how many plants were sampled per field, per season etc? How many fields were sampled each year? How were the plants sampled? Randomly?
3) This is of personal interest – I wondered why aphids are preserved in a 90% ethanol and 75% lactic acid mix, and parasitoids in a 96% ethanol solution?
4) Where are the percentages provided in Figure 2? Should they be provided like in Figure 1? Or are they approximated from the size of the balls in the figure?
5) I wonder if figure 5 could be nice as a line graph showing aphid numbers with time? It would be easier to get an idea of changes in insect numbers and composition.
6) You state here that although Blackman and Dixon report that only 3% of aphids are anholocyclic, all 25 species in your study were found to be anholocyclic. The sentence is worded in such a way to imply that this is an unexpected finding. This is surely not unexpected or unusual, particularly when there are over 4,000 species of aphids worldwide. In a warm Mediterranean climate where conditions are favourable year-round, we would expect anholocycly to prevail. However, as we move into more temperate and polar climates, we would expect the level of holocycly to increase to enable aphids to survive colder winters. Since the likes of Myzus persicae, Aphis gossypii, Rhopalosiphum padi (3 of the most common species found in your study) have a global distribution and display a sexual lifecycle in cold climates, I suspect that the majority, if not all, of your 25 aphids species fit in to the 3% reported by Blackman and Dixon and that the fact that these species display an anholocyclic lifecycle in your Mediterranean study is completely normal. I would suggest rewording this part to avoid sensationalising an expected finding.
Author Response
Dear referee,
Thank you for your editing and comments. We did all your corrections to the text. The text was edited for English again. Material and method was improved.
Aphids have very soft body and long term preservation in lactic acid + alcohol is better than only alcohol.
The last paragraph of the discussion has been rewritten in line with your suggestion.
Best regards
